# The Impact of HPP-Assisted Biocontrol Approach on the Bacterial Communities’ Dynamics and Quality Parameters of a Fermented Meat Sausage Model

**DOI:** 10.3390/biology12091212

**Published:** 2023-09-06

**Authors:** Norton Komora, Cláudia Maciel, Joana Isidro, Carlos A. Pinto, Gianuario Fortunato, Jorge M. A. Saraiva, Paula Teixeira

**Affiliations:** 1CBQF—Centro de Biotecnologia e Química Fina—Laboratório Associado, Escola Superior de Biotecnologia, Universidade Católica Portuguesa, Rua Diogo Botelho 1327, 4169-005 Porto, Portugal; nortonsnk@hotmail.com (N.K.); nuario@hotmail.it (G.F.); 2Genomics and Bioinformatics Unit, Department of Infectious Diseases, National Institute of Health Dr. Ricardo Jorge, 1649-016 Lisbon, Portugal; joana.isidro@insa.min-saude.pt; 3LAQV-REQUIMTE, Chemistry Department, University of Aveiro, Campus Universitário de Santiago, 3810-193 Aveiro, Portugal; carlospinto@ua.pt (C.A.P.); jorgesaraiva@ua.pt (J.M.A.S.)

**Keywords:** fermented meat sausage, clean label, microbial communities, high-pressure processing, bacteriophage, lactic acid bacteria, pediocin PA-1

## Abstract

**Simple Summary:**

The consumption and appreciation of traditional foods have grown worldwide in the last century, valuing their cultural heritage, natural-associated content, and manufacturing process. Nonetheless, the increase in foodborne outbreaks associated with the consumption of these types of foods has become a concern. High-pressure processing (HPP)-assisted biocontrol is a non-thermal, multi-hurdle technology based on a cocktail of environmentally friendly antimicrobials that can be applied in synergy with mild HPP at low temperatures (≤10 °C) to promote the inactivation of target pathogenic bacteria and provide a sustainable alternative to conventional thermal food decontamination. Beyond the preservation of the physicochemical and organoleptic properties of foods, HPP-assisted biocontrol scarcely influences the endogenous microbiota.

**Abstract:**

Traditional foods are increasingly valued by consumers, whose attention and purchase willingness are highly influenced by other claims such as ‘natural’, ‘sustainable’, and ‘clean label’. The purpose of the present study was to evaluate the impact of a novel non-thermal food processing method (i.e., HPP-assisted biocontrol combining mild high hydrostatic pressure, listeriophage Listex, and pediocin PA-1 producing *Pediococcus acidilactici*) on the succession of bacterial communities and quality of a fermented sausage model. A comparative analysis of instrumental color, texture, and lipid peroxidation revealed no significant differences (*p* > 0.05) in these quality parameters between non- and minimally processed fermented sausages throughout 60-day refrigerated storage (4 °C). The microbiota dynamics of biotreated and untreated fermented sausages were assessed by 16S rRNA next-generation sequencing, and the alpha and beta diversity analyses revealed no dissimilarity in the structure and composition of the bacterial communities over the analyzed period. The innovative multi-hurdle technology proposed herein holds valuable potential for the manufacture of traditional fermented sausages while preserving their unique intrinsic characteristics.

## 1. Introduction

The slow food movement encompasses the ongoing global demand for locally and artisanally produced foods. This trend is mainly supported by consumers who seek to reduce their ecological footprint and eliminate highly industrialized food products from the supply chain (i.e., food products with a significant content of chemical preservatives, which allow an extended shelf life) [1]. The natural content of traditional food products (TFPs) is positively correlated with consumers’ intentions to purchase and consume them [2]. Furthermore, the promotion of TFP has emerged in the last century and plays an important role in authenticating an ethnic or national culture [3]. For example, the institutionalized protection of certain foods with national identity has developed exponentially, especially in Europe, where TFPs are part of the European culture, identity, and heritage [4]. In a European cross-cultural consumer-driven study, the four main dimensions for the concept of TFP were identified as habitat (nature), place of origin, processing, and sensory characteristics [5]. In addition, processing technology and novelty change were identified as the main dimensions for consumer expectations of TFP innovation [5].

In this sense, the emergence of antimicrobials with “clean label” status (e.g., (bacterio)phages, bacteriocins, and essential oils) as well as innovative food processing technologies (e.g., high-pressure processing (HPP), pulsed electric field, and cold plasma, among others) seem to align with the goal of improving TFP since previous studies in this field of research have shown no or low effects on their physicochemical and organoleptic characteristics [6,7]. In particular, HPP, bacteriophages, and bacteriocinogenic lactic acid bacteria (LAB) have recently been explored as innovative hurdles applied in the multi-hurdle technology concept. Although positive interactions between phages and bacteriocinogenic LAB for a bactericidal effect on a target pathogen have been documented, bacterial regrowth still occurs in biocontrol carried out under atmospheric conditions [8,9]. High hydrostatic pressure (HHP) is a well-established non-thermal technology in which a packaged food product is immersed in a pressurized fluid, such as water, at pressures up to 600 MPa, which promotes microbial inactivation by disrupting cellular integrity [10]. Komora et al. [11] demonstrated that the simultaneous application of mild HPP with phage Listex and pediocin PA-1 improved the antimicrobial efficacy of those agents through sensitization of HPP-treated cells, resulting in higher pathogen inactivation and no regrowth.

*Alheira* is a traditional spontaneously fermented sausage produced in the Northern region of Portugal. It is manufactured with a mixture of meats (pork and chicken, among others), bread, and olive oil. Elected as one of the seven Portuguese gastronomic wonders, *Alheira de Mirandela*, along with *Alheira de Barroso—Montalegre*, and *Alheira de Vinhais,* are the sausages with Protected Geographical Indication (PGI) status currently commercialized in Portugal. Despite the cultural and anthropological aspects of *Alheira* production and consumption, there is a significant issue regarding *Listeria monocytogenes* since a considerable number of medium-scale industrially produced *Alheira* batches have shown counts above 100 CFU g^−1^ [12].

Foodborne outbreaks and recalls involving ethnic/traditional foods are increasing worldwide. *Listeria monocytogenes* and *Salmonella* spp. have been identified as the principal foodborne pathogens associated with these events [13]. As a result, there is an urgent need for risk assessment and mitigation strategies to ensure food safety.

The objective of this study was to evaluate the impact of HPP-assisted biocontrol (i.e., combination of phage Listex and pediocin PA-1 producing *Pediococcus acidilactici* assisted by mild HPP) on the quality characteristics and bacterial communities’ dynamics of a fermented sausage model (*Alheira*) paste throughout 60 days of refrigerated storage.

## 2. Materials and Methods

### 2.1. Microorganisms and Inoculum Preparation

*Pediococcus acidilactici* strain HA-6111-2, an autochthonous lactic acid bacterium (LAB) producing pediocin PA-1 and isolated from Alheira [14], was chosen as the protective culture. The baroresistance of this strain, along with the impact of high-pressure processing (HPP) on bacteriocin synthesis following pressure exposure, has been previously validated [15]. The bacterium was preserved at −20 °C in de Man, Rogosa, and Sharpe (MRS) broth (Lab M, Lancashire, UK) supplemented with 30% (*v v*^−1^) of glycerol (Sigma-Aldrich, Steinheim, Germany). The inoculum was prepared as previously described by Komora and Maciel et al. [16]. A former study addressing the impact of HPP on the phage Listex stability in several food matrices (including *Alheira*) was conducted by Komora et al. [17]. PhageGuard Listex (based on listeriophage P100) stock solution (Micreos Food Safety, Wageningen, The Netherlands) was stored at 4 °C in the original buffer, and the initial concentration was 10^11^ plaque-forming units (PFUs) mL^−1^. A working solution of phage (*ca.* 10^10^ PFU mL^−1^) was freshly prepared on the day of the experiment as previously described by Komora and Maciel et al. [16].

### 2.2. Preparation of Alheira Paste

*Alheira* sausage was selected based on its relevance among Portuguese traditional foods and on the prevalence of *L. monocytogenes* in this type of food matrix (fermented meat sausage). The listericidal effect of the HPP-assisted biocontrol in *Alheira* was previously documented by Komora and Maciel et al. [16]. Fresh samples of *Alheira* within no more than one week from the production date (as estimated by the expiration date) were purchased from a local supermarket (Porto, Portugal) and transported to the laboratory in insulated boxes containing ice packs. The *Alheira* paste (*ca.* 200 g) was aseptically removed from the casings and placed in sterile stomacher bags. One set of samples was maintained at atmospheric pressure (0.1 MPa) under refrigeration (4 °C) (non-processed control samples), while the other set was inoculated with 0.1% (*v w*^−1^) of phage Listex and 0.1% (*v w*^−1^) of *P. acidilactici* HA-6111-2. Homogenization was manually performed by gently massaging the sample (*ca*. 3 min). The final phage concentration reached 10^8^ PFU g^−1^ and *P. acidilactici* achieved a final level of 10^7^ colony forming units (CFUs) g^−1^ in *Alheira* paste. This set of samples was submitted to mild HPP (300 MPa, 5 min, 10 °C) and afterwards stored at 4 °C.

### 2.3. High Hydrostatic Pressure Treatments

The samples prepared as described in Section 2.2 were double vacuum sealed in low-permeability polyamide-polyethylene bags (PA/PE-90, Albipack—Packaging Solutions, Águeda, Portugal) and then pressurized at 300 MPa (10 °C) for 5 min in a hydrostatic press from Hiperbaric 55 (Burgos, Spain). The pressurization rate was established at ca. 100 MPa per 7 s, and water was used as the pressure-transmitting fluid. Samples were treated in three independent batches.

### 2.4. Quality Parameters of Fermented Meat Sausage over Refrigerated Storage

*Alheira* samples were evaluated at predetermined intervals (0, 15, 30, 45, and 60 days) during their refrigerated storage at 4 °C in order to determine the viability of the proposed multi-hurdle technology. Both unprocessed and minimally processed samples were analyzed for physicochemical characteristics (color, lipid peroxidation) and texture (consistency/firmness).

#### 2.4.1. Color Analysis

Color parameters were determined with a digital Minolta colorimeter (Model CR-400, Konica Minolta Camera Ltda, Tokyo, Japan) by conducting five measurements in distinct parts of *Alheira* paste for each treatment. The Commision Internationale de L’éclairage (CIE) parameters lightness (L*), redness/greenness (a*), and yellowness/blueness (b*) were determined. Additionally, the hue angle (h°), the chroma or saturation index (*C*_ab_), and total color differences (ΔE) were calculated according to Cruxen et al. [18] and Tomasevic et al. [19].

#### 2.4.2. Texture Analysis—Compression/Extrusion Tests

Compression/extrusion tests were conducted using a texture analyzer TA.XT Plus (Stable Micro Systems, Surrey, UK) with a 5 kg load cell and a 5 mm cylindrical plunger.

The force was exerted at 50% deformation and at a constant penetration speed of 2 mm min^−1^. Three penetrations were performed for each treatment, equilibrating sausage pastes at room temperature (20 °C) as described by Ben Slima et al. [6]. Results were expressed as maximum force (extrusion force, N g^−1^), which provides an indication of the sample’s consistency/firmness.

#### 2.4.3. Thiobarbituric Acid Reactive Substances (TBARS)

Lipid peroxidation was evaluated using the TBARS method as described by Arslan and Soyer [20], with minor modifications. Briefly, 5 g of samples were dissolved in 10 mL of trichloroacetic acid (TCA biochemical/PanReac Applichem, Darmstadt, Germany) and homogenized using a BagMixer for 3 min. Subsequently, the extracts were centrifuged (5000× *g* for 10 min at 4 °C), and the interphases were collected and carefully transferred to new tubes. Then, 9 mL of TCA-2-thiobarbituric acid (TBA, Merck, Darmstadt, Germany) reactive solution was added to each tube, which was vortexed for 15 s. The resulting mixture was boiled for 40 min, cooled to room temperature, and the absorbance measured at a wavelength of 530 nm. The calibration curve was previously prepared using the 1,1,3,3-tetraethoxypropane (TEP, Sigma-Aldrich) standard. The results (TBARS concentration) were expressed as mg malondialdehyde (MDA) per kg of sample.

### 2.5. Bacterial Communities’ Dynamics of Fermented Meat Sausage Model

At pre-set time intervals (0, 15, 30, 45, and 60 days), non- and minimally processed fermented sausage samples were analyzed with regard to the bacterial diversity through a culture-independent approach, namely next-generation sequencing (NGS) of the V3-V4 region of 16S rRNA.

#### 2.5.1. Total DNA Extraction Procedure

Total DNA extraction from fermented sausage model samples was performed according to Chaillou et al. [21] and Quijada et al. [22], with some modifications. Briefly, 5–10 g of homogenized paste of *Alheira* were mixed with the appropriate volume (dilution 1:10) of previously warmed at 45 °C sterile saline buffer (composed by 0.1% (*w v*^−1^) tryptone (Biokar Diagnostics, Beauvais, France), 0.85% (*w v*^−1^) NaCl (Sigma-Aldrich), and 1% (*w v*^−1^) Tween 80 (Sigma-Aldrich), and homogenized for 5 min in a BagMixer at maximum speed (stomacher). Then, the mixture was centrifuged (600× *g*, for 10 min, at 4 °C) and the interphase was carefully transferred to a new tube. The interphase was centrifuged (7000× *g*, for 10 min, at 4 °C), to recover bacterial cells, and the pellet was carefully washed twice with 1 mL of sterile 0.1 M PBS (VWR Chemicals, Radnor, PA, USA) and used for DNA extraction by using the DNeasy^®^ mericon^®^ Food Kit (Qiagen, Hilden, Germany) following manufacturer instructions. DNA estimated concentration and quality (absorbance ratio 260 and 280 nm) in the extracts were evaluated using the NanoDrop One (ThermoScientific, Wilmington, DE, USA) and by electrophoresis on 0.8% (*w v*^−1^) agarose gel. Total DNA (from each treatment and sampling day) used for further NGS represents a pool from the three independent replicates (i.e., three different batches) as previously described [23].

#### 2.5.2. 16S rRNA Gene Amplicon Sequencing and Library Preparation

Microbial diversity was assessed by NGS of the V3-V4 hypervariable region of the 16S rRNA gene using conditions previously described by Macieira et al. [24]. All 16S rRNA data were analyzed with Kraken v1, following the parameters formerly described [24]. At each taxonomic rank, only the taxa representing ≥0.1% of the classified reads (with ≥10 reads) in at least one of the samples were considered. The relative abundance, the alpha diversity indices (Shannon–Wiener and Simpson), and the beta diversity metrics (Bray–Curtis dissimilarity matrix) were determined for each sample. A graphical representation of the relative abundance of the bacterial groups (with an occurrence higher than 0.3%) at the family and genus taxonomic levels was performed using the R package ggplot2 [25].

### 2.6. Statistical Analysis

The differences in quality parameters were analyzed using the one-way analysis of variance (ANOVA) with Tukey’s post hoc test (SPSS, Version 23.0, Inc., Chicago, IL, USA), when homogeneity of variance was assumed. Statistical analysis of biodiversity and ecological communities was performed with the Vegan R [26] and R Stats packages.

## 3. Results and Discussion

### 3.1. Instrumental Color Evaluation of Non- and HPP-Phage-LAB Treated Sausages

The evolution of the instrumental color parameters for non- and minimally processed (HPP, Listex, and LAB) fermented sausage paste over the 60-day refrigerated storage period is presented in Table 1.

No significant differences (*p* > 0.05) were observed between control and minimally processed samples with respect to the instrumental color coordinates L*, a*, and b*, over the sampling period. Moreover, the color stability was also evaluated concerning the CIELCh values (color saturation (*C*_ab_) and color tone (h°)) and total color difference (ΔE), and it was found that these parameters were not significantly (*p* > 0.05) altered in the minimally processed samples when compared to the control samples.

The correlations between the variations of the different quality attributes over the 60-day storage period were depicted by Pearson’s correlation coefficient heatmap (Figure 1). According to Pearson’s analysis, a positive correlation was found between instrumental yellow (b*) and chromatic tonality (h°), or color intensity (C), while a negative correlation was found between b* (yellowness) and a* (redness), regardless of whether fermented sausages were minimally processed or not. This latter correlation (enhancement of yellowness concurrent with a decrease in redness) may be ascribed to a slight lipid oxidation.

These results were in close agreement with previously documented effects of HHP and *Pediococcus* species on the color attributes of meat products. For instance, no significant differences in the color characteristics during shelf-life testing of HPP-treated Iberian dry-cured *Salchichón* and *Chorizo* were observed after processing at 500 MPa (5 min, 18 °C) and 600 MPa (8 min, 16 °C), respectively [27,28]. High-pressure-treated Iberian dry-cured ham (200 or 300 MPa, 15 or 30 min, 14 °C) presented slight differences in redness (CIE a*) only perceived immediately after processing, while a short storage time decreased the differences to similar CIE color values of control samples (0.1 MPa) [29]. With respect to *Pediococcus* species as protective cultures and their impact on color attributes, Ben Slima et al. [6] reported no effect on color parameters of beef fermented sausage after the incorporation of *P. acidilactici* MA 18/5 M. Van Ba et al. [30] also described a minor impact on CIE color values of pork fermented sausage inoculated with *P. acidilactici* SA20 (Starterkulturen Almi 20).

### 3.2. Minimal Processing Impact on Alheira Texture: Compression/Extrusion Tests

Extrusion force (EF), conventionally defined as the maximum force in kg attained during extrusion, was determined in *Alheira* samples throughout the shelf life, and values ranged from 0.61 to 0.68 N g^−1^. No significant differences (*p* > 0.05) between control and HPP-Listex-LAB-treated samples were observed (Table 1). Similar results have been previously reported by Rigdon et al. [31], who performed the texture profile analysis (TPA) of beef fermented summer sausage (inoculated with a *P. acidilactici* culture) and documented no impact of HHP (586 MPa, 150 or 300 s, 4 °C) on the TPA profile of pressurized samples. The addition of *P. acidilactici* SA20 also did not impact the TPA properties of pork fermented sausage [30].

On the other hand, other authors described significantly affected TPA traits as a consequence of the incorporation of protective cultures; Casquete et al. [32] described higher hardness, gumminess, and chewiness values in Spanish dry-cured fermented sausage inoculated with autochthonous *P. acidilactici* MC184 and *Staphylococcus vitulus* RS34 in comparison with the commercial culture batch. Correspondingly, Wang et al. [33] also found higher TPA values for hardness and springiness in an inoculated Chinese fermented sausage (with *Pediococcus pentosaceus*, *Latilactobacillus sakei,* and *Staphylococcus xylosus*) in comparison with its control (spontaneous fermentation).

Overall, the differences in the TPA profile elicited by the addition of protective cultures could be due to different enzymatic/metabolic activity of the incorporated bacteria since the acidifying capacity of LAB and acid-induced protein denaturation could lead to an increased firmness of sausages [6].

### 3.3. Lipid Peroxidation Evaluation through Thiobarbituric Acid Reactive Substances (TBARS) Assays

The determination of TBARS as an indicator of lipid peroxidation in *Alheira* was performed, and the concentrations ranged from 0.87 ± 0.07 to 1.41 ± 0.03 mg malondialdehyde per kg (Table 1). In general, *Alheira* presented low TBARS values, and no significant differences (*p* > 0.05) were observed between HPP-phage-LAB-treated and control samples. Until day 15 of storage, TBARS values were within the threshold level of 1 mg MDA kg^−1^ established by Ripoll et al. [34] for meat product rancidity and also described as the spoiled limit by the Chinese standard (GB-T 9959. 2-2008) [35]. However, many other authors support an acceptable limit of 2 mg MDA kg^−1^ [36,37], in which case *Alheira* samples are in conformity during the entire storage period.

According to the Pearson’s correlation analysis (Figure 1), TBARS value variation presented positive correlations with total color difference and instrumental yellow (b*), while a negative correlation was found with redness, irrespective of the samples processing. The positive correlation between TBARS and yellowness may be ascribed to the onset of lipid oxidation, which may confer an intensification of the yellowish-brownish color of *Alheira*. This finding corroborates the previously hypothesized negative relationship between instrumental a* and b*. Moreover, those color alterations may explain the positive correlation observed between TBARS and total color difference.

Pressure magnitudes exceeding 300 MPa have been suggested as the limit to trigger the lipid oxidation intensification in high-pressure processing, besides the holding time, the temperature inside the vessel, and the fat content/type (i.e., the degree of saturation) of the food matrix, which modulate the extension of lipid oxidation; generally, the higher the pressure level and holding time, the higher the lipid oxidation/TBARS value [38]. Accordingly, Liu et al. [39] investigated the combined effect of enterocin LM-2 and HHP (200 or 400 MPa, 10 min, 17 °C) on sliced ham quality attributes and described no differences in TBARS values between samples treated at 200 MPa and the control (0.1 MPa), whereas 400 MPa led to an increase in the oxidative process.

In a study performed on *Salchichón* dry-cured fermented sausage, the addition of *P. acidilactici* SP979 demonstrated an antioxidant effect in TBARS values compared to the naturally fermented counterpart [40]. However, Van Ba et al. [30] investigated the lipid stability of pork fermented sausage inoculated with different commercial cultures and found no significant differences in TBARS values between the batch with added *P. acidilactici* and the control (endogenous fermentation). Similarly, Casquete et al. [32] documented no significant differences in TBARS values of *Salchichón* produced with commercial cultures or autochthonous *P. acidilactici* MC184 and *S*. *vitulus* RS34.

It should be highlighted that the assessment of a single strain cannot rule out information regarding the biochemical/technological transformation properties of different bioprotective cultures on fat stability. This is because different metabolic pathways can be observed within the same species due to strain diversity.

### 3.4. Impact of the Minimal Processing on Alheira Microbiota Dynamics Assessed by 16S rRNA Diversity Analysis

The hypervariable V3/V4 region of the 16S rRNA gene was amplified from the total genomic DNA of *Alheira* paste over 60 days of storage to unravel the impact of HPP-assisted biocontrol on the structure and composition of bacterial communities. Briefly, a total of 716,885 sequences (raw reads) were obtained from 10 samples (non- and minimally processed fermented meat sausage models at 0, 15, 30, 45, and 60 days), and an average of 99.3% (98.4–99.7%) of the sequencing reads (after quality filtering) were classified by Kraken as bacterial reads.

A detailed taxonomic analysis identified 35 distinct bacterial families in total, of which only the 13 families with an occurrence higher than 0.3% were represented (Figure 2). *Leuconostocaceae* was the major component of the microbiota during the early stages of the storage period in both analyzed samples, and its relative abundance continuously decreased from the fifteenth day onwards.

Contrastingly, from the thirtieth day of refrigerated storage until the endpoint, *Lactobacillaceae* became the dominant family (75.9 and 91.3% relative abundance values at the sixtieth day in minimally processed and non-processed samples, respectively). The background microbiota was predominantly composed of non-LAB meat spoilage bacteria belonging to the class *Gammaproteobacteria* (*Pseudomonadaceae*, *Moraxellaceae*, *Morganellaceae*, *Erwiniaceae*, and *Shewanellaceae* families), whose relative abundance also decreased concomitantly with the emergence of *Lactobacillaceae*. In consonance with these findings, the development and succession of the *Lactobacillaceae* family through the progression of storage time of cured/fermented meat products have been extensively described since these bacteria are well-known to adapt and evolve in acidic environments [41,42].

In congruency with these fluctuations in the bacterial community structure, the analysis of the intra-sample α-diversity metrics unveiled a decrease in the Shannon–Wiener (H’) (Figure 3) and Simpson (D) indices in minimally processed (H’: 1.89 to 0.74; D: 0.76 to 0.38) and control samples (H’: 1.89 to 0.47; D: 0.72 to 0.16) over the analyzed period. Moreover, the Shannon–Wiener indices did not evidence significant differences (*p*-value = 0.6299) in the bacterial diversity (richness) between the two samples (mean Shannon index values of 1.25 and 1.08 for processed and control samples, respectively).

The bacterial community profiles of the non-processed and minimally processed samples were examined based on beta-diversity dispersion analysis through principal-coordinate analysis (PCoA) using the Bray–Curtis dissimilarity matrix (Figure 4). The analysis revealed that the minimally processed samples presented a similar taxonomic structure compared with the non-processed samples. The organization of the PCoA plot highlighted the lack of dissimilar segregation between the microbial communities, corroborating the scarce impact of the novel technology proposed herein on the taxonomic profile of the minimally processed fermented sausage model.

At the genus level (Figure 5), *Weisella* species were predominant during the initial stages of storage, being surpassed by *Lactobacillus* species on the thirtieth day of storage, which was consolidated as the most abundant family of the microbiota until the endpoint (relative abundance values of 57.8% and 62% for non- and minimally processed samples, respectively).

Other genera from the order *Lactobacillales* present in considerable percentages were *Leuconostoc* and *Pediococcus*; as expected, *Pediococcus* emerged as one of the dominant species through intermediate and terminal periods of shelf-life, being the second and third most abundant genus at the sixtieth day of storage in control (relative abundance value of 33.3%) and HPP-Listex-LAB (relative abundance value of 13.5%) samples, respectively. *Acinetobacter* and *Pseudomonas* from the *Pseudomonadales* order and *Providencia* from the *Enterobacteriaceae* family were the most prevalent non-LAB spoilage bacteria in the initial phase of storage, followed by *Brochothrix*, *Bacillus*, *Psychrobacter*, *Shewanella,* and the *Anoxibacillus* genus.

The results presented herein are in close accordance with the microbial communities’ patterns previously described in *Alheira* as well as the microbial shifts triggered by the innovative antimicrobial hurdles. *Alheira* bacterial community profiles were first documented by Albano et al. [43] utilizing conventional cultural methods and PCR-denaturing gradient gel electrophoresis (DGGE) and band sequencing; according to cultural methods, LAB were the dominant microbiota. Fingerprints from DGGE analysis demonstrated that the order *Lactobacillales* was predominant, whereas bacteria from the order *Bacillales* and class *Gammaproteobacteria* represented most of the remaining microbial communities. Macieira et al. [24] determined the impact of *Lactiplantibacillus plantarum* bacteriocinogenic strain ST153Ch, as a starter culture, on the microbiota of *Alheira de Vitela* by applying NGS tools; *Leuconostocaceae* and *Lactobacillaceae* were the most abundant families in all analyzed samples, and the addition of *Lpb. plantarum* ST153Ch during fermentation increased the relative abundance of the genus *Lactobacillus* over the genus *Leuconostoc* in treated samples compared to counterpart samples (spontaneous fermented). Furthermore, a significant impact on the microbial diversity parameter and an accentuated decrease in *Weisella* species were observed with the incorporation of bacteriocinogenic culture.

The impact of innovative hurdles (e.g., HPP, essential oils, and bacteriocins) intended for food processing on the microbial communities of several food matrices such as fish, soy sauce mash, Iberian dry-cured ham, soybean paste, and fermented sausages has been documented [7,42,44,45]. In opposition to the results observed in the present study, a huge impact on microbiota diversity exerted by the bacteriocinogenic *Lpb. plantarum* LP-1 incorporation in low-salt sausages was observed [42], and an even more pronounced effect on the bacterial ecology of fish fillets was achieved by applying thymol and enterocin AS-48 in activated film packaging combined with HHP [7].

Overall, HPP-assisted biocontrol appears to have a minimal impact on the fermented sausage model microbiota dynamics. In fact, slight differences in the relative abundance were observed in unprocessed and minimally processed samples, albeit the main trends in the bacterial communities over the 60 days of storage were followed by both sets of samples. Further studies comprising the addition of these antimicrobials at the beginning of the production of fermented meat sausages should be performed in order to evaluate the effect of the decontamination process on fermentation dynamics during maturation.

## 4. Conclusions

The impact of the HPP-assisted biocontrol on the quality attributes and bacterial communities of the fermented meat sausage model was documented, and no significant differences between unprocessed and minimally processed samples were obtained concerning texture, color, and lipid peroxidation, while minor changes were observed regarding the ecological diversity of the microbiota. Hence, in the current work, the microbial community profiling, coupled with the physicochemical data, was pivotal in demonstrating that the novel non-thermal decontamination technology proposed herein has a high potential to preserve the unique traits of traditional food products (TFP).

## Figures and Tables

**Figure 1 biology-12-01212-f001:**
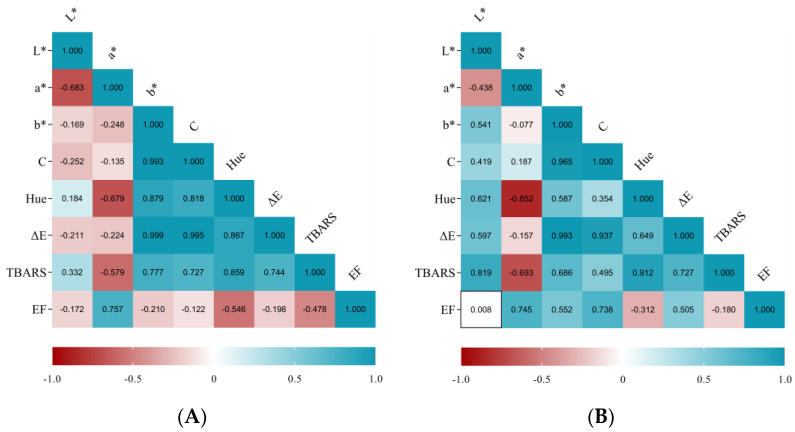
Heatmap of correlation analysis between quality attributes of non-processed (**A**) and minimally processed (**B**) fermented sausages based on Pearson’s correlation coefficients.

**Figure 2 biology-12-01212-f002:**
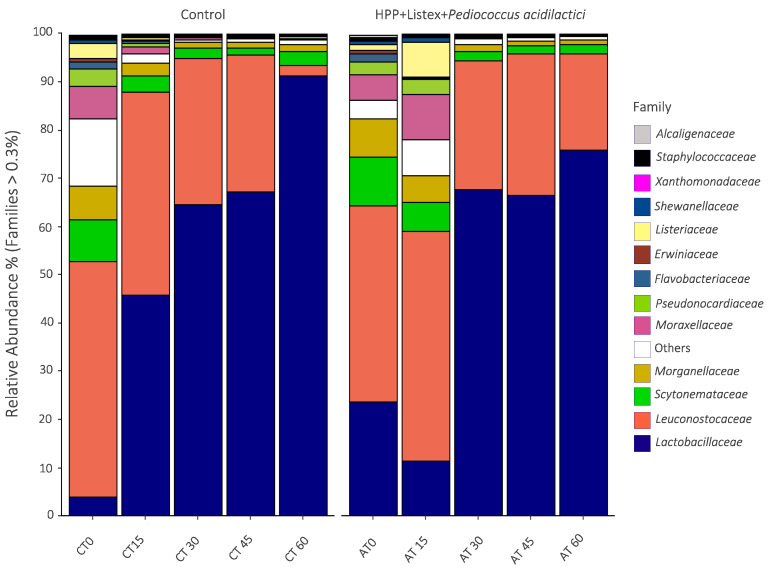
Relative abundance of bacterial communities at the taxonomic level of family in the non-processed (control) and minimally processed (HPP, Listex, and *P. acidilactici*) fermented meat sausage model over the 60 days of refrigerated storage (CT-control, AT-after treatment). The relative abundance was expressed as the percentage of the families with an occurrence higher than 0.3% in at least one sample.

**Figure 3 biology-12-01212-f003:**
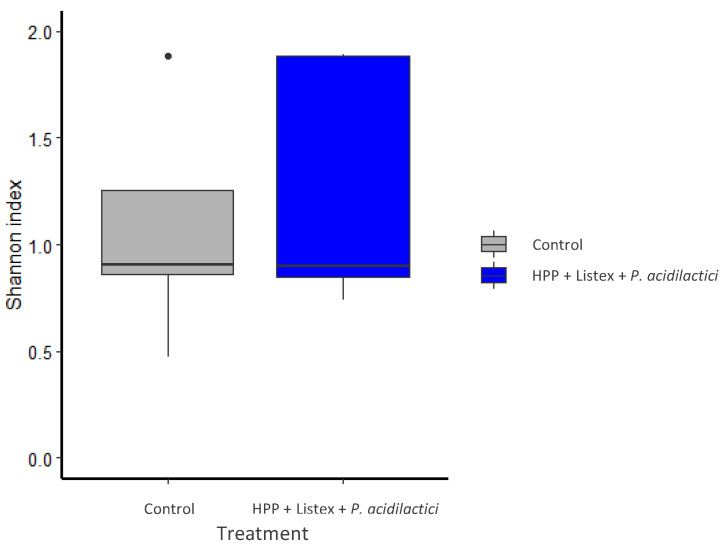
Impact of minimal processing on the alpha diversity of bacterial communities in fermented meat sausage (Shannon–Wiener Diversity Index). The gray box plot corresponds to non-processed (control) and the blue box to minimally processed (HPP, Listex, and *P. acidilactici*) fermented sausages. The circle indicates an outlier.

**Figure 4 biology-12-01212-f004:**
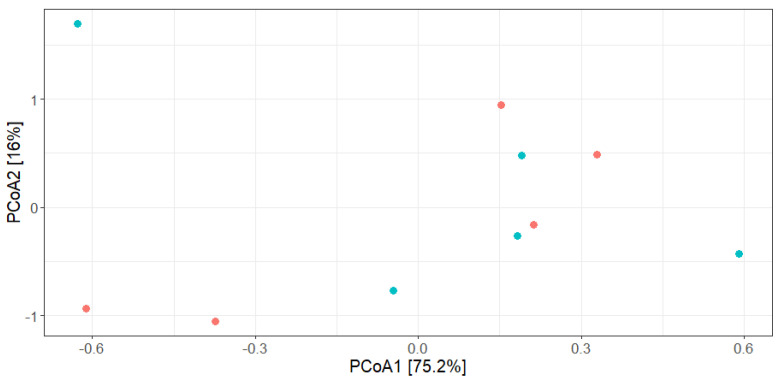
Principal Coordinate Analysis (PCoA) is based on a Bray–Curtis dissimilarity matrix determined at the family level. The plot represents the distribution of the different samples over the 60 days of storage period (blue circle, non-processed; orange circle, minimally processed samples).

**Figure 5 biology-12-01212-f005:**
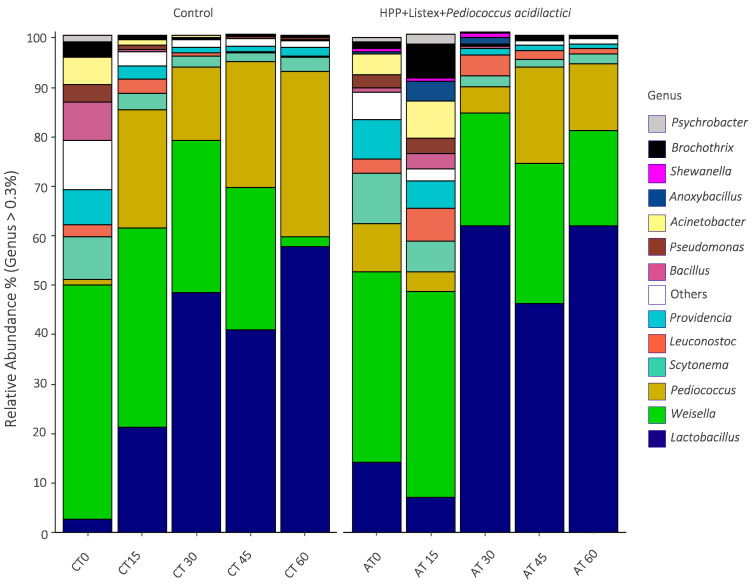
Relative abundance of bacterial communities at the taxonomic level of genus in non-processed (control) and minimally processed (HPP, Listex, and *P. acidilactici*) fermented meat sausage models throughout the 60 days of refrigerated storage (CT-control, AT-after treatment). The relative abundance was expressed as the percentage of the genera with an occurrence higher than 0.3% in at least one sample.

**Table 1 biology-12-01212-t001:** Evolution of quality parameters of non- and minimally processed fermented sausages over 60 days of storage.

Sampling Time (Days)		Color Parameters						TBARS (mg kg^−1^)	Extrusion Force (N g^−1^)
		Lightness (L*)	Redness (a*)	Yellowness (b*)	Hue Angle (h°)	Chroma (*C*_ab_)	ΔE		
0	C	59.60 ± 0.84	13.15 ± 0.94	28.49 ± 2.99	65.17 ± 0.82	31.38 ± 3.10	0	0.89 ± 0.09	0.65 ± 0.02
	HPP+Listex+LAB	58.38 ± 0.82	13.24 ± 0.54	29.85 ± 2.53	66.03 ± 0.95	32.66 ± 2.53	0	0.87 ± 0.07	0.61 ± 0.02
15	C	57.82 ± 0.73	13.82 ± 0.89	32.72 ± 2.13	67.09 ± 0.65	35.52 ± 2.27	4.68 ± 2.29	0.96 ± 0.10	0.66 ± 0.09
	HPP+Listex+LAB	58.39 ± 1.13	14.43 ± 0.67	33.51 ± 2.02	66.69 ± 0.43	36.48 ± 2.11	3.93 ± 2.19	0.95 ± 0.05	0.68 ± 0.01
30	C	58.53 ± 0.32	12.57 ± 1.10	32.22 ± 0.75	68.71 ± 1.26	34.59 ± 1.08	4.05 ± 0.58	1.11 ± 0.06	0.62 ± 0.04
	HPP+Listex+LAB	58.23 ± 0.41	11.85 ± 0.29	32.10 ± 0.57	69.74 ± 0.53	34.22 ± 0.56	2.69 ± 0.49	1.09 ± 0.04	0.61 ± 0.02
45	C	59.91 ± 0.65	12.22 ± 1.13	33.22 ± 2.89	69.81 ± 0.22	35.39 ± 3.10	5.04 ± 2.63	1.17 ± 0.03	0.64 ± 0.01
	HPP+Listex+LAB	59.56 ± 1.21	12.28 ± 1.08	33.15 ± 2.95	69.68 ± 0.02	35.35 ± 3.14	4.09 ± 2.44	1.23 ± 0.05	0.64 ± 0.04
60	C	59.43 ± 1.24	12.82 ± 1.16	33.78 ± 1.47	69.21 ± 1.94	36.15 ± 1.43	5.49 ± 1.43	1.34 ± 0.02	0.64 ± 0.05
	HPP+Listex+LAB	59.41 ± 1.26	12.01 ± 0.98	33.82 ± 1.43	70.47 ± 0.70	35.89 ± 1.68	4.47 ± 1.44	1.41 ± 0.03	0.62 ± 0.04

C—control; ΔE—total color difference; TBARS—thiobarbituric acid reactive substances.

## Data Availability

Not applicable.

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
