# Peer review of "The Impact of HPP-Assisted Biocontrol Approach on the Bacterial Communities’ Dynamics and Quality Parameters of a Fermented Meat Sausage Model"

_biology, 2023, doi:10.3390/biology12091212_

Round 1

Reviewer 1 Report

This study focused on the impact of HPP-assisted biocontrol approach on the bacterial communities’ dynamics and quality parameters of a fermented meat sausage model. The manuscript was well written. However, there are also sever questions that might be the reviewer’ attention.

1. The high-pressure processing can also influence the listeriophage Listex and Pediococcus acidilactici ?

2. The theme of this manuscript was high-pressure processing, a contrasted biocontrol approach without high-pressure processing might be need to highlight the importance of HPP.

3. Figure 1, the color of this picture was too dark to display the number.

4. Based on the results of this manuscript, HPP-assisted biocontrol appeared to have a scarce impact on the fermented sausage model microbiota dynamics. However, what was the beneficial effect of HPP ?

no

Author Response

Dear Editor and Reviewers,

We would like to thank the opportunity to submit a revised version of the manuscript entitled “The impact of HPP-assisted biocontrol approach on the bacterial communities’ dynamics and quality parameters of a fermented meat sausage model”. We are grateful for your insightful comments and valuable suggestions, which we have carefully considered. Hence, we have made the necessary revisions to address each reviewer's concerns in order to improve the clarity and rigor of our study. Please find below our detailed responses (written in blue) to the reviewers' comments (written in black), where we have provided point-by-point responses and outlined the specific changes made in the revised manuscript. In this new version of the manuscript all modified parts from the original version were highlighted in yellow.

Reviewer 1 Comments

This study focused on the impact of HPP-assisted biocontrol approach on the bacterial communities’ dynamics and quality parameters of a fermented meat sausage model. The manuscript was well written. However, there are also sever questions that might be the reviewer’ attention.

We are grateful for the positive feedback, insightful comments and valuable suggestions kindly provided by the reviewer 1, which have certainly improved our manuscript. Please find below our detailed responses (written in blue) to the reviewer’s questions (written in black).

Point 1: The high-pressure processing can also influence the listeriophage Listex and Pediococcus acidilactici ?

Response to Point 1:

The authors thank the reviewer’s question. As stated in lines 96-97 and lines 101-102, the feasibility of applying HPP to these microorganisms, maintaining their biological activity has been previously evaluated, confirmed and documented.

Ln 96-97 “the baroresistance of P. acidilactici as well as the impact of HPP on the bacteriocin synthesis following pressure exposure were previously validated [15].”

Ln 101-102 “A former study addressing the impact of HPP on the phage Listex stability in several food matrices (including Alheira) was conducted by Komora et al. [17].”

Point 2:  The theme of this manuscript was high-pressure processing, a contrasted biocontrol approach without high-pressure processing might be need to highlight the importance of HPP.

Response to Point 2: While we appreciate the reviewer’s comment, we must clarify that the subject of this manuscript was the effect/impact of a new non-thermal technology on the quality parameters of a fermented sausage model. In fact, biocontrol without the aid of high pressure has already been tested [16], and it was found that the decontamination of Listeria was not as significant and immediate as when the biocontrol was carried out with high pressure. Nonetheless, in order to address the reviewer’s raised concern, the introduction has been revised (lines 62-73) and, in addition to mentioning cases of interaction between phages and bacteriocinogenic lactic acid bacteria with HPP (lines 64-47), there is a new sentence clarifying the choice of this combination of hurdles (lines 70-73).

Point 3: Figure 1, the color of this picture was too dark to display the number.

Response to Point 3: The authors thank the reviewer’s suggestion. Figure 1 was revised accordingly.

Point 4: Based on the results of this manuscript, HPP-assisted biocontrol appeared to have a scarce impact on the fermented sausage model microbiota dynamics. However, what was the beneficial effect of HPP ?

Response to Point 4:  The authors acknowledge the reviewer’s comment. We must clarify that the effect of high pressure was not individually evaluated in this manuscript, as the present minimal non-thermal treatment was designed by combining the three hurdles (phage, bacteriocinogenic lactic acid bacteria and high pressure). More information on the relevance of high pressure to the effectiveness of the treatment can be found in our previous work, which demonstrated the advantage of applying high pressure in synergy with biocontrol compared to biocontrol under atmospheric conditions [16]. Nevertheless, the introduction section has been updated to clarify the choice of this combination of hurdles.

Reviewer 2 Report

Some of the required corrections are included in attached file.

There were no distinct advantages of using the HPP-assisted biocontrol approach compared to control. In both untreated and treated samples the counts of listeria sp. decreased significantly (fig. 2) over the storage period.

Main drawbacks in the work:

Authors didn't check for or quantify pathogenic listeria sp that are more likely to present (according to authors) in the purchased Alheira.

Anti-listerial effect of HPP-assisted biocontrol approach was not experimented in this study, just quoted a reference.

Minor editing of English language required

Author Response

Dear Editor and Reviewers,

We would like to thank the opportunity to submit a revised version of the manuscript entitled “The impact of HPP-assisted biocontrol approach on the bacterial communities’ dynamics and quality parameters of a fermented meat sausage model”. We are grateful for your insightful comments and valuable suggestions, which we have carefully considered. Hence, we have made the necessary revisions to address each reviewer's concerns in order to improve the clarity and rigor of our study. Please find below our detailed responses (written in blue) to the reviewers' comments (written in black), where we have provided point-by-point responses and outlined the specific changes made in the revised manuscript. In this new version of the manuscript all modified parts from the original version were highlighted in yellow.

Reviewer 2 Comments

Some of the required corrections are included in attached file.

The authors thank the reviewer’s comments and suggestions. The suggestions included in the attached file were addressed in the new version of the manuscript and the English grammar was finely revised.

Point 1: There were no distinct advantages of using the HPP-assisted biocontrol approach compared to control. In both untreated and treated samples the counts of listeria sp. decreased significantly (fig. 2) over the storage period.

Response to Point 1:

 The authors thank the reviewer’s comment. Figure 2 provides an overview of microorganisms at the family taxonomic level. The Listeriaceae family encompasses two genera: Brochothrix and Listeria, the latter of which currently comprises 26 valid published species. Therefore, the figure refers not only to the presence of Listeria sp. but also to Brochothrix, which is commonly known as a spoilage microorganism in meat. Nevertheless, the listericidal effect of this technology has been duly tested and previously published by our group [16]. Since bacteriophage P100 and bacteriocin pediocin PA-1 are narrow-spectrum antimicrobials with specific activity against Listeria monocytogenes, no differences are expected at the family level.

Point 2: Authors didn't check for or quantify pathogenic listeria sp that are more likely to present (according to authors) in the purchased Alheira.

Response to Point 2:

The authors acknowledge the reviewer’s comment. The absence of Listeria monocytogenes in the purchased Alheira was confirmed according to the ISO 11290-1:2017.

Point 3: Anti-listerial effect of HPP-assisted biocontrol approach was not experimented in this study, just quoted a reference.

Response to Point 3: The authors acknowledge the reviewer’s comment. A previous study focused on the development of a new non-thermal decontamination process was conducted by our group, in which the hurdles were evaluated individually and in combination, and it was found that the association of phage, bacteriocinogenic LAB and HPP was the best combination for immediate inactivation of L. monocytogenes. The present study focuses on evaluating the impact of the multi-hurdle technology (previously validated concerning decontamination of Listeria monocytogenes, in the cited reference 16) on the evolution of the microbiota and the quality attributes of the food matrix, namely color, lipid oxidation, and texture, since Alheira is a fermented food.

Reviewer 3 Report

The manuscript is well written and organized. The methods were adequate to the objective of the work.

In order to contextualize efficiently the reader in the introduction it could be added some details about the applicability of phage Listex and pediocin PA-1 producing Pediococcus acidilactici on food processing, if possible isolated and in combination.

In the last sentence of conclusion the authors mentioned that the applied technology present a high potential to be implemente in the decontamination, but the work described did not asses the decontamination of the product, only assess some quality parameters and the microbiota profile. This sentence should be rephrased to refer only directly to the conclusions of the work.

Author Response

Dear Editor and Reviewers,

We would like to thank the opportunity to submit a revised version of the manuscript entitled “The impact of HPP-assisted biocontrol approach on the bacterial communities’ dynamics and quality parameters of a fermented meat sausage model”. We are grateful for your insightful comments and valuable suggestions, which we have carefully considered. Hence, we have made the necessary revisions to address each reviewer's concerns in order to improve the clarity and rigor of our study. Please find below our detailed responses (written in blue) to the reviewers' comments (written in black), where we have provided point-by-point responses and outlined the specific changes made in the revised manuscript. In this new version of the manuscript all modified parts from the original version were highlighted in yellow.

Reviewer 3 Comments

The manuscript is well written and organized. The methods were adequate to the objective of the work.

We are grateful for the positive feedback, insightful comments and valuable suggestions kindly provided by the reviewer 3, which have certainly improved our manuscript.

Point 1:  In order to contextualize efficiently the reader in the introduction it could be added some details about the applicability of phage Listex and pediocin PA-1 producing Pediococcus acidilactici on food processing, if possible isolated and in combination.

Response to Point 1:

The authors thank the reviewer for pointing out the importance of including a brief contextualization of the applicability of biocontrol agents on food processing. Following the reviewer's suggestion, the introduction has been revised accordingly.

Point 2:  In the last sentence of conclusion the authors mentioned that the applied technology present a high potential to be implemented in the decontamination, but the work described did not assess the decontamination of the product, only assess some quality parameters and the microbiota profile. This sentence should be rephrased to refer only directly to the conclusions of the work.

Response to Point 2:

The authors thank the reviewer comment. Hence, in order to address the abovementioned issue, the Conclusion section was revised accordingly (lines 437-440).

Round 2

Reviewer 2 Report

The manuscript has been improved. However, there is still room for improvement in the discussion section